# Current Advancements in Addressing Key Challenges of Therapeutic Antibody Design, Manufacture, and Formulation

**DOI:** 10.3390/antib8020036

**Published:** 2019-06-03

**Authors:** Vicki Sifniotis, Esteban Cruz, Barbaros Eroglu, Veysel Kayser

**Affiliations:** School of Pharmacy, Faculty of Medicine and Health, The University of Sydney, Sydney 2006, Australia; vsif0221@uni.sydney.edu.au (V.S.); ecru7298@uni.sydney.edu.au (E.C.); barbaros.eroglu@sydney.edu.au (B.E.)

**Keywords:** therapeutic antibody, stability, aggregation, manufacture challenges, formulation

## Abstract

Therapeutic antibody technology heavily dominates the biologics market and continues to present as a significant industrial interest in developing novel and improved antibody treatment strategies. Many noteworthy advancements in the last decades have propelled the success of antibody development; however, there are still opportunities for improvement. In considering such interest to develop antibody therapies, this review summarizes the array of challenges and considerations faced in the design, manufacture, and formulation of therapeutic antibodies, such as stability, bioavailability and immunological engagement. We discuss the advancement of technologies that address these challenges, highlighting key antibody engineered formats that have been adapted. Furthermore, we examine the implication of novel formulation technologies such as nanocarrier delivery systems for the potential to formulate for pulmonary delivery. Finally, we comprehensively discuss developments in computational approaches for the strategic design of antibodies with modulated functions.

## 1. Introduction

Since the first therapeutic monoclonal antibody (mAb) Orthoclone OKT3^®^ (Janssen Biotech, Horsham, PA, USA) was approved by the USA Food and Drug Administration in 1986, whole antibody therapeutics have become and persistently remain the most dominant and significant biologic therapeutic platform in the pharmaceutical industry [1,2]. To date, therapeutic antibodies treat a plethora of indications including cancers, infections, autoimmune disorders, and cardiovascular and neurological diseases [3]. The whole antibody therapeutics platform is regarded as the most promising class of pharmaceutical technology to date; it is continually being applied to newly identified biological targets and implemented in many formats to produce strategically engineered next generation antibody therapeutics, otherwise termed “biobetters” [4,5,6,7,8,9]. The international ImMunoGeneTics information system^®^ (IMGT^®^, Montpellier, France) database reveals that as of December 2018, 65 whole antibodies and 18 next generation fragment or recombinant fusion antibody-based therapies are approved for clinical use, with hundreds more in clinical trials expected to reach market.

Whole therapeutic mAbs are presently the dominant antibody platform approved for clinical use (Figure 1), although antibody engineering technologies have advanced in recent years to produce highly optimized, strategically engineered biobetter therapies, along with biosimilar mAbs reaching market to compete against their originator. Further whole mAb formats include antibody–drug conjugates (ADCs), bispecifics, isotype-switched, and glycoengineered. These additional formats have been strategically designed to introduce exceptional potency, to engage dual biological targets, and to modulate Fc effector functions. Fragments of mAbs such as the crystallizable fragment (Fc), antigen binding fragment (Fab), and single-chain variable fragment (scFv) possess key functions such as specificity to a biological target or immunological activation. The isolation of these fragments for fusion with other mAb fragments, biologically functional proteins, cytotoxic drugs, or drug carriers has been the crux of ingenuity in developing the next generation of biobetter therapies [4,5,6,7,8,9,10,11,12,13,14,15]. Figure 2 depicts several examples of prominent biobetter formats, providing a general representation of current fragment mAbs, whole mAb bispecifics, fragment mAb multispecifics, and fragment mAb fusion therapeutics.

## 2. Overview of mAb Production Challenges and Considerations

Whole therapeutic mAbs require a mammalian expression system to produce the biologically functional product; however, a mAb fragment and recombinant fusion mAb products with simplified (or lacking) glycosylation are suitable for lower organism expression platforms [16]. Unlike oligopeptides, which can be chemically synthesized, whole therapeutic mAbs are considerably larger (with monomer ranging from 140–160 kDa) and comprise of four peptide chains (two heavy and two light chains) bound together by disulfide bonds and interchain non-covalent interactions. Further to this, antibodies contain glycosylation in a conserved region of the Fc (N297) that contributes to its stability and immune effector functions [17,18]. Aside from peptide synthesis, cell machinery is required to glycosylate, fold, orient, and covalently bind the antibody peptide chains in order to produce the complete, biologically functional antibody product. Manufacturing biologically functional whole mAb product is therefore commercially unfeasible through chemical synthesis and insufficient in lower organism expression platforms such as bacteria, yeast, insect, and plant cells that may not have the machinery to produce the equivalent tertiary structure and glycosylation profiles. In particular, many industrially relevant bacterial strains such as *E. coli* are completely deficient in the machinery to add post-translational glycosylations; yeasts hyper-mannosylate glycans, which cause immunogenicity; and insect cells which are deficient in sialylation machinery and produce immunogenic glycan structures [16]. A secretion of the mAb product for purification is suboptimal for several lower organism expression platforms such as *E. coli* due to poor productivity and harsh culture conditions that promote product degradation. Protein is therefore produced intracellularly, as inclusion bodies and harvest involves further processing steps such as cell lysis, inclusion body recovery, protein solubilization, and renaturation prior to further downstream purification steps [19,20]. Despite these pitfalls, the development of lower organism expression systems is of high commercial interest due to the simplified culture conditions, cheaper media requirements, rapid organism growth, and higher product yield as compared to mammalian expression systems [16,21].

In considering the requirements through the entire process of mAb discovery, manufacture, formulation, and disease treatment, several key challenges arise which have sparked overwhelming interest in pursuit of achieving better mAb manufacturing outcomes and treatment strategies. As with all biotherapeutics, mAbs and mAb-based therapeutics are limited to production in cell-based expression systems, which is considerably costly and inefficient, can have varied yields depending on the product and expression system, and requires downstream processing to remove biological contaminants introduced from the expression system. Despite affinity chromatography being a robust technology for the initial capture of a mAb for purification, the capture process and further downstream processes such as viral inactivation applies the mAb product to harsh pH and salt conditions, which can chemically degrade the mAb, leading to product instability and loss [5,22,23].

Many factors through the manufacture process influence glycosylation and charge heterogeneity of mAbs, which affects their biophysical and pharmacological properties. Though not specifically discussed in this review, the improvement and control mAb production technologies address these variations to reduce formulation heterogeneity and off-target cytotoxicities.

A common challenge, as seen with all biotherapeutics, is that mAbs and mAb-based therapeutics are currently restricted to lyophilised and liquid-based formulations for intravenous (IV) or subcutaneous (SC) delivery to achieve maximum bioavailability. Protein self-association and intrinsic stability drive this limitation, in that viscosity and propensity to aggregate are dependent on mAb concentration. Formulations are optimized to achieve the highest dosing concentration at the minimum achievable volume for injection, without compromising the quality of the mAb in formulation [24]. Viscosity remains a key limiting factor for formulating as a SC administration—certain mAb therapies are suitable and others not based on their solubility, self-association, and aggregation profiles. Alternative non-invasive administration strategies such as pulmonary delivery causes additional mechanical stress that further contribute to mAb instability and loss. Furthermore, oral delivery is unsuitable due to chemical and enzymatic degradation, as well as poor absorption in the gastric and intestinal environments [5,25,26,27,28,29].

A brief overview of considerations through the different concept stages of therapeutic mAb development is depicted in Figure 3. The main challenges and considerations in the manufacture and formulation of mAb therapeutics are briefly summarized in Table 1.

## 3. mAb Discovery and Manufacture Technologies

Traditional technology for whole therapeutic antibody discovery required the immunization of animals—primarily mice with a target antigen for the generation of a mixed population of B-lymphocytes-producing antibody against the target. The B-lymphocytes would be isolated for immortalization to produce monoclonal hybridoma cell lines that secrete antibody candidates of interest for further panning through display technologies to isolate potential leads [5]. This method has led to the generation of highly specific mAb libraries of non-human origin that were potentially immunogenic and less efficacious at eliciting an immune response as compared to wholly human mAbs. Technologies arose to address immunogenicity by producing chimeric and humanized antibodies through the grafting of the variable domains (chimeric) or complementarity-determining region (CDR) residues (humanized) isolated from lead non-human antibodies to a human antibody framework. To further improve the humanization strategy, transgenic mice were developed with their murine antibody heavy and light chain genes replaced with equivalent human genes; they consequently expressed wholly human antibodies for discovery [4,5,6].

Phage display technology remains the most prominent selection technology for panning antibody gene pools for specificity to a target antigen, as it effectively couples the expression of mAb proteins to the genes that encode them for the panning of high affinity leads. Antibody variable genes from B-lymphocyte pools are isolated through polymerase chain reaction (PCR), and cloned into a phage expression vector that presents the expressed mAb on the surface of the phage, the library of vectors is rescued in phage, and the phage library is screened against a specific target antigen. Phage pools undergo several rounds of selection to isolate high affinity candidates, the variable genes of the lead mAb candidates can then be isolated and sequenced for the design of a mAb drug format and the development of a suitable expression platform [5,47]. Error-prone PCR-based mutagenesis has propelled display technology for the generation of enormous libraries for target affinity screening. Yeast surface display further improves these screening technologies, as glycosylation sites introduced in CDRs are expressed through this platform [5,48].

The expression, yield, and quality of a mAb can vary quite substantially in hybridoma cell lines. High yielding mammalian expression systems have been developed to meet the commercial need for high expression efficiency, scalability, quality, and reproducibility. Antibody genes of interest are introduced into suitable expression vectors and transfected into highly efficient mammalian cell lines for antibody expression and secretion; a mAb can then be directly captured from the culture supernatant through affinity chromatography. Currently, the majority of mammalian expression systems for commercial whole therapeutic antibody expression are based on stable chinese hamster ovary (CHO), mouse myeloma (NS0), and mouse hybridoma (Sp2/0) cell lines [4,5]. However, in the context of research and development, the transient expression in human cell lines such as embryonic kidney (HEK 293), amniotic (CAP), a hybrid of HEK 293 and lymphoma (HKB-11), and embryonic retina (PER.C6) are favored over stable CHO expression due to the ease and speed of production of workable quantities of antibody for preliminary studies [16,30,31,32,49,50]. The generated mAb library undergoes relevant in vitro testing along with formulation stability screening to exclude candidates that have poor manufacturability attributes [51]. Lead mAb candidates that show potential for further investigation would then be developed for high yielding stable expression and more rigorous characterization leading up to therapeutic development. However, a drawback from changing from human to hamster expression systems is the difference in glycosylation profile. CHO expressed mAbs produce immunogenic non-human glycoform Neu5Gc and have a higher composition of sialylation, which may result in reduced antibody-dependent cellular cytotoxicity (ADCC). Therefore in the early stages of development, leads need to be identified and thoroughly characterized in the expression system to be developed for commercial manufacture before committing to industrial scale up [16,30,32]. Amongst other biotherapeutics, approved mAb Fc fusion therapeutics dulglutide (Trulicity^®^, Eli Lilly, Indianapolis, IN USA), efmoroctocog alpha and eftrenonacog alpha (Eloctate^®^ and Alprolix^®^, Biogen, Cambridge, MA, USA), are manufactured in a HEK 293 expression system, which is giving rise to the acceptance of human-based expression systems for the production of mAb-based therapeutics [4]. However, HEK 293 expression systems are prone to inducing mAb aggregation in cultures, which is detrimental to cell viability and creates a loss of product during manufacture. Hence, HKB-11 and PER.C6 are the preferred commercial human cell lines for mAb expression in human cell lines, specifically [16,31,32].

Established lower organism expression systems for fragment or recombinant fusion mAb therapeutics include bacteria such as *E. coli*, yeasts such as *S. cerevisiae* and *P. pastoris*, plants such as tobacco, algae, and insects such as silkworm [5,16,52,53,54]. Expression platforms that use *E. coli* in particular are considered high risk due to the potential endotoxin contamination in the mAb product, of which complete endotoxin removal requires further purification steps [19]. However, several approved mAb fragment- and recombinant-based therapies are produced in an *E. coli*-based expression system, including pegol conjugated Fab’ certolizumab pegol (Cimzia^®^, Celltech UCB, Brussels, Belgium), Fab ranibizumab (Lucentis^®^, Genentech, San Francisco, CA, USA), and recombinant Fc fusion romiplostim (Nplate^®^, Amgen, Thousand Oaks, CA, USA). Lower organism expression platforms such as *E. coli* and *P. pastoris* continue to be the preferred option for the manufacture of fragment mAb formats due to the relative ease, high yield, and reduced cost for manufacture as compared to mammalian expression platforms.

An emerging in vitro cell-free synthesis technology is being developed with bacterial and CHO cell lysates which have the potential to alleviate formation of undesirable biological byproducts, as the machinery from the cell lysates purely express protein from the mAb genes that are introduced to the system. Though this technology is not currently applicable for industrial scale manufacture, it holds much promise as an alternative to live culture. For instance, culture maintenance and highly defined media are no longer necessary, reactions can run continuously, lysates can be recycled with reproducible results, unmodified linear DNA is suitable for the system (thus alleviating a need for multiple cloning steps), and additional enzymes can be introduced to the system for the engineering of specific post-translational modifications [33,34,55,56,57,58]. Despite the apparent advantages of this technology for mAb manufacture, cell lysates encounter the same challenges as their host cell expression system. That is, post-translational modifications of mAb product from lysates from lower level organisms are limited by the endogenous machinery of that host organism. Furthermore, preparation of mammalian cell lysates is challenging, and yield produced from these lysates is considerably low [59,60].

Current bioprocess engineering and cell line development strategies have been crucial in enhancing the manufacturability of mAbs. Many current mAb therapeutic expression platforms make use of CHO cell lines that have been commercially developed with dihydrofolate reductase or glutamine synthetase deficiency for enhanced stability selection through resistance to methotrexate and methionine sulfoximine inhibition, respectively [50,61]. Mammalian cell lines have been adapted to suspension culture, as they support higher cell densities and mAb titers in the absence of serum from the culture media, for large scale fed-batches, perfusion systems, or continuous culture systems. Further to this, cell lines are continually being engineered for enhanced metabolic functions, introducing glycosylation pathways, superior secretion, and resistance to apoptosis for prolonged survival, in efforts to generate super-producer cell lines that can sustain a continuous culture [62,63,64,65,66,67,68].

Gene and expression vector design and development are specifically tailored to the expression system to maximize mAb expression and cell line stabilization through host cell codon optimization and the addition of highly efficient transcription, secretion, selection, and integration elements [69,70,71,72,73]. Vectors can contain a single site for gene insertion for the subsequent co-transfection of vectors that carry the antibody heavy and light chains, or both chains can be cloned into a dual expression vector. In more elaborately designed recombinant mAb drug formats that require the expression of several genes, vector technologies have implemented multicistronic expression to enhance the efficiency of the vector system. The stable or transient transfection of vectors is performed on highly viable cell cultures with high cell density. Antibody heavy and light chain ratios may require adjustment in transfection for optimizing expression and secretion. A reporter vector that expresses green fluorescent protein is typically included to observe transfection efficiency during optimization [74,75]. Liposome-mediated transfection is preferential over all other mammalian transfection strategies and is induced chemically through the complexing of DNA to a cationic lipid prior to or during addition to a culture. Polyethylenimine is the most prevalently used transfection reagent despite the development of superior reagents such as Lipofectamine™ 2000 and Freestyle™ MAX (Thermo Fisher Scientific, Waltham, MA, USA), LyoVec™ (InvivoGen, San Diego, CA, USA), FuGENE 6™ (Promega, Madison, WI, USA), and *Trans*IT-PRO^®^ (Mirus, Madison, WI, USA), owing to its lower cost and relative efficiency [73,75,76,77,78].

The commercial manufacture of mAbs has transitioned to serum-free, chemically defined, and animal-free media which has removed a source of expression variability and has been driven particularly from the advent of mad cow disease [50,79]. Several media supplements such as plant and yeast digests (peptones/hydrolysates), surfactants (e.g., Pluronic F-68), DNA methyltransferase (azacytidine), and histone deacetylase (sodium butyrate, valproic acid) inhibitors have been found to support cell viability and enhance mAb expression [79,80,81,82,83,84,85]. Mild hypothermia of the culture has also demonstrated to reduce cell expansion, support prolonged cell viability, and enhance mAb expression [73,86,87,88,89,90]. Further to this, continuous supplementation in expression cultures of uridine, manganese chloride, and galactose demonstrated successful the glycan engineering of expressed mAb, where mAb hyper-galactosylation was promoted to enhance complement-dependent cytotoxicity (CDC) activity [91].

In the manufacturing process of fed-batch, perfusion, and continuous culture systems, media is continuously fed in the culture system, whilst parameters such as cell viability, cell density, and metabolite levels are monitored in real time until the parameters indicate that it is the optimal time to harvest the expressed mAb from the culture supernatant. In perfusion and continuous culture systems, a feed is removed from the culture simultaneously to the media addition, the difference being that the cell dilution rate in continuous culture is optimized to remain equal or higher than the cell growth rate, which allows the perpetuation of mAb expression and requires the continuous harvest of the supernatant [92,93,94]. The harvest of the supernatant requires clarification technologies such as the use of precipitants/flocculants (e.g., polyethylene glycol, diethylaminoethyl dextran, caprylic acid, and polyethylenimine), high throughput centrifugation, and filtration methods to remove cells and biological debris, for the lowering the loading burden in the lead up to mAb capture through affinity chromatography [19,95,96,97,98,99,100]. The mechanical stresses from centrifugation and filtration are unavoidable, although they can induce mild mAb product loss through fragmentation and aggregation.

Protein A-ligand-based affinity chromatography is the most robust, efficient, and prevalently used mAb capture technology, owing to its selectivity and high affinity to various human Fc, as well as its efficient dissociation of captured mAbs at a low pH for reuse. Fragment mAbs and recombinant formats based on Fabs and single chain variable fragments (scFv) are unable to take advantage of protein A affinity capture, and alternatives have been developed, such as capturing proteins G, M, and L, which bind at different mAb epitopes; ion exchange chromatography; and polyhistidine tagged capture through immobilized metal chromatography [100,101,102]. A further advantage of affinity chromatography in manufacture is the integration of a viral inactivation step by holding the low pH-eluted mAb product prior to further purification steps; however, this applies the mAb product to low pH at high concentrations, which can induce mild mAb instability, aggregation, and the formation of acidic variants [5,22,100,103,104].

Post mAb capture, further polishing steps are required to remove further contaminants such as host cell proteins and DNA, as well as leached affinity chromatography ligand- and mAb-degradation products such as fragments, aggregates, and ionic variants. The polishing steps are designed based on the specific properties of the mAb product, such as the isoelectric point and molecular weight (MW), to implement appropriate chromatographic technologies such as anion and cation exchanges, hydrophobic interaction, and multimodal and size exclusion that will separate the mAb product from the manufacture-introduced impurities [22,100,103]. The additional chromatographic steps unavoidably apply the mAb product to buffers of varying ionic strength and high concentrations, which can again induce mild mAb instability and aggregation. Post polishing steps, the purified mAb product undergoes a further viral removal step, such as filtration, and then it undergoes buffer exchange and concentration through ultrafiltration or diafiltration methods to prepare the bulk mAb product for formulation [100,105].

## 4. Formulation Strategies and Considerations

Strategies to improve the formulation of mAbs and mAb-based therapies continues to be an ongoing challenge, as is faced with all biotherapeutics. Firstly, whole therapeutic mAbs are unsuitable for non-invasive oral, nasal, or pulmonary routes of administration as they are susceptible to chemical and enzymatic degradation in the gastrointestinal tract. The bioavailability of mAbs through these routes is poor, owing to mAbs’ polar surface charge and relatively large MW, limiting transport through mucosal membranes.

Fragment mAb platforms, together with excipient and PEGylation technologies, have been developed to help circumvent transportation limitations for pulmonary delivery, specifically. However, physical stresses applied to the mAb (e.g., shear stresses from the aerosolization of liquid formulations for a pressurized meter dose and a nebulization delivery, or from the production of dry powder for inhalation) pose further challenges of mAb instability, leading to degradation and reduced efficacy. [8,26,28]. Few biologics have been successfully approved for pulmonary delivery, most notably insulin formulation Afrezza^®^ (MannKind, Westlake Village, CA, USA), and, currently, an erythropoietin-Fc fusion and a nanobody-targeting respiratory syncytial virus are in clinical trials, showing promise for the further development of this administration strategy for both the systemic and localized delivery of mAb-based therapeutics [5,28].

Several drug carrier technologies such as microencapsulation, liposome, and nanoparticle formulations inherently enhance the stability and control the release of mAbs, which can prolong their half-life. These nanocarrier formulation strategies are of intense interest, as they hold promise for developing less invasive inhaled formulations of mAb-based therapeutics with superior attributes to currently established formulations [41,106,107,108,109].

The majority of currently approved mAb therapeutics are formulated for IV, although commercial interest has directed technologies to develop injectable mAb formulations which has seen success in many approved mAb therapies thus far, primarily SC for systemic delivery, along with a few intravitreal and intramuscular formulations for tissue-specific indications. Amongst others, anti-HER2 antibody trastuzumab was originally developed as an IV formulation and was successfully repurposed as a SC formulation [110,111,112], whereas next generation therapies have directly moved towards SC formulation, such as with anti-TNF-α antibody adalimumab [113]. Injectable administrations offer several advantages over IV administration, especially in the treatment of chronic diseases in regards to a reduced burden to allied health services, patient tolerance, and adherence to treatment; however, the intrinsic physicochemical properties of mAbs may be undesirable for injectable formulation. In considering the low volume, injection pressure, and the typically high (>100 mg) effective dose of mAb for injectable delivery, the viscosity and aggregation propensity of mAbs become key formulation challenges to address, as they are dependent on mAb concentration. Certain mAb therapies are suitable, and others are not based on their solubility, viscosity, self-association, intrinsic stability, aggregation, and precipitation profiles.

Injectable mAb formulations can be further improved with the use of excipients to increase solubility, reduce viscosity, and enhance the stability of mAbs. Excipients are considered for a formulation based on their physicochemical properties, pharmacokinetics, and safety. For example, polysorbates are commonly used in biologics as a stabilizing agent; however, their addition in high concentrations can denature proteins and cause adverse side effects such as injection site reactions [114,115,116]. Injectable mAb formulations are co-formulated with recombinant human hyaluronidase, specifically as a permeation enhancer for more efficient absorption into tissue, although the inclusion of this additional biologic adds further burden to the formulation’s viscosity and propensity to aggregate [5,8,25,26,27,28,117,118,119,120,121,122,123]. Antibody therapies for IV administration are prepared as lyophilised powder for reconstitution and further dilution, and injectable administrations are prepared as liquid-based formulations in pre-filled syringes. Liquid formulations of mAbs are more susceptible to physiochemical degradation, are less stable, and have a reduced shelf-life as compared to lyophilised formulations. However, drying technologies to produce lyophilised formulations apply the mAb to physical stresses that induce instability and degradation, leading to reduced efficacy [124,125,126].

Long-term stability predictions are elucidated from formulation screening in accelerated aggregation studies, as part of the preliminary screening process of generated mAb libraries [127]. The stability profiles of mAbs can be greatly affected by the amino acid composition, structure, and potential glycosylation in their variable region CDRs that are isolated through the screening and maturation process. Elucidating the causes for reduced solubility or increased aggregation propensity has to be considered together with the molecular interactions between the mAb and the biological target as to not compromise affinity. Computational tools have been developed to elucidate amino acids within the mAb CDR structure that have a high propensity to aggregate, and strategies for improving solubility and aggregation profiles have been developed through direct amino acid substitutions and the strategic addition of glycosylation sites [128,129,130,131,132]. The characterization of mAb stability and target interaction for optimization is considered a fundamental step as part of preliminary drug discovery, as the applicability for development, manufacture, and formulation of the mAb therapeutic is governed by the mAbs stability profile.

## 5. Improving mAb Tissue Penetration for Cancer Treatment

Antibody therapies are currently restricted to invasive parenteral routes of administration for maximum bioavailability and systemic distribution; however, delivery to the specific target tissue, such as tumors for cancer treatments, continues to be a challenge. Penetration to tissue from blood vessels is again poor, owing to mAbs’ polar surface charge and relatively large MW, limiting transport through physiological barriers. The efficiency of tissue penetration is further influenced by systemic and local mAb clearance rates. Enhancement to mAbs’ affinity to their biological target through maturation and strategic mutation technologies have led to faster diffusion rates of mAb to target for increased efficacy. However, for tumourous tissue specifically, penetration through tumors is restricted by the tumor’s binding site barrier, in which antigens expressed on the tumors periphery capture the majority of the mAb released to surrounding tissue; this effect is exaggerated with overexpressed antigen. The enhancement of whole mAbs’ affinity beyond 1 nM has specifically shown that there is no further improvement of tumor diffusion rates, tissue penetration, and accumulation [42,43,133]. An increased affinity of mAbs to cellular targets also leads to the increased uptake, internalization, and catabolism of mAbs, which reduces ADCC and increases the mAb clearance rate. This mechanism is exploited through the development of high affinity ADCs to target the delivery and release of potent drugs, inducing targeted cell death [43,44,45].

Fragment mAb platforms have shown better tissue penetration and biodistribution than whole mAb therapeutics; however, a pitfall of smaller peptides lacking an Fc region is a highly reduced in vivo half-life and poor retention times. Technologies to improve the half-life of mAb fragments have been primarily through PEGylation, along with strategic Fc mutation and glycosylation engineering to enhance neonatal Fc receptor (FcRn) recycling. Furthermore, hyper-glycosylation technology has been successfully applied in other biotherapeutics, such as with glycosylated erythropoietin Aranesp^®^ (Amgen, Thousand Oaks, CA, USA). Conversely, nanocarrier platforms that are relatively larger as compared to whole mAbs have demonstrated superior pharmacokinetics and tumor retention, as well as previously mentioned enhanced stability and the sustained, controlled release of mAbs [41]. These enhanced properties have generated considerable interest in developing the systemic delivery of mAb-nanoparticle platforms for superior tumor penetration, as well as targeted and sustained therapeutic drug delivery. Further to nanocarriers, additional formulation strategies developed for the controlled release of mAbs include hydrogels and crystalline antibodies, which have shown success as stable, injectable formulations for development [5,8,15,41,42,43,134,135,136].

In addition, to further enhance the stability and half-life of other biotherapeutics, recombinant technologies allowed their fusion to mAb Fc as to introduce FcRn recycling as a protection mechanism, which has innovatively expanded the applicability of mAb-based therapeutic platforms. Notable examples include the TNF Receptor–Fc fusion protein Enbrel^®^ (Amgen, Thousand Oaks, CA, USA), which acts as an inhibitor to overexpressed TNF-α in autoimmune diseases, and the Factor IX–Fc fusion protein Alprolix^®^ (Biogen, Cambridge, MA, USA), which is a blood factor supplement for hemophiliacs [11,46,133].

## 6. Strategic Modulation of mAb Immune Effector Functions

The modulation of mAbs effector functions through isotype switching, glycoengineering, and strategic mutations have proved advantageous for the development of more effective mAb treatment strategies. Antibody binding to Cq1 promotes the complement cascade, Fc_γ_R1A/B, Fc_γ_R2A, and Fc_γ_R3A/B receptors to activate immune effector functions; Fc_γ_R2B counter-balances the effector response, and FcRn prolongs mAb half-life. Binding to these receptors is primarily done through sites in the C_hinge_, C_H_2, and the conserved glycosylation region in the Fc (N297).

IgG3 does not efficiently bind to FcRn, which reduces its half-life to approximately seven days, as opposed to a 21 day half-life for the other IgG isotypes. In most instances of mAb design, an extended half-life is preferred as to prolong the effective dose of a mAb in serum [15]. IgG2 and IgG4 mAbs have a reduced effector function as compared to IgG1 and are thus used in instances where minimal engagement to the immune system is warranted to increase safety of the mAb therapy—with ADCs reducing off-target cytotoxicity, for instance. Eculizumab (Soliris^®^, Alexion Pharmaceuticals, New Haven, CT, USA) is the first (and thus far only) approved recombinant IgGκ 2(C_H_1/C_hinge_)–4(C_H_2/C_H_3) hybrid whole mAb. Further cross-sub-class variant mAb therapies are in development, with key amino acid substitutions from the IgG2 and IgG4 sub-classes introduced for intentionally suppressed effector functions [4,13]. The removal of the conserved glycosylation site through amino acid substitution of N297 or T299 (aglycosylation) has also demonstrated reduced effector function; however, this happens at the expense of mAb stability and is therefore not a suitable strategy for whole mAb therapeutics [11,14,15,137,138,139,140].

On the other hand, enhancing ADCC and CDC is useful for engaging the immune system to tumor tissues in the absence of a conjugated cytotoxic drug. Certain glycoengineered modifications to the conserved glycosylation region in the Fc, such as deficiency in core fucose (afucosylation) and hyper-galactosylation, have demonstrated enhanced mAb binding to Fc_γ_R3A and Cq1, specifically for an enhanced ADCC and CDC effect [13,43]. Afucosylated mAbs benralizumab (Fasenra™, AstraZeneca, London, UK) and mogamulizumab (Poteligeo^®^, Kyowa Hakko Kirin, Tokyo, Japan), as well as low fucose content mAb obinutuzumab (Gazyva^®^, Genentech, San Francisco, CA, USA), are currently approved therapies, with several further in development (along with clinical trials), which demonstrates the commercial interest and applicability of this technology in improving mAb therapeutics through enhancing ADCC. However, the significance of manipulating sialylation in mAb therapeutics is a controversial topic, with several reports demonstrating a reduction in ADCC and CDC and others reporting no observable difference. This highlights a clear need to characterize and define the in vivo efficacy of such manipulation [11,13,14,43].

No currently approved mAb therapies contain amino acid substitutions for an improved half-life through modulated binding properties to FcRn, improved CDC through binding to complement factor Cq1, or improved ADCC through enhanced binding affinity to Fc_γ_R1A and 3A. However, several mutations have been reported, patented, and are in clinical development, demonstrating the commercial interest in this technology for improving mAb therapeutics [6,11,13,15,133,137,140,141,142,143].

For a more comprehensive understanding of mAb engineering strategies for modulated immune effector functions, we direct the reader to the following reviews [13,14,15,140].

## 7. Computational Approaches for Aggregation Prediction and Rational Design of mAbs

The advancement of in silico analysis of mAb peptide sequences, structures, conformation, and their associated biological interaction has been integral to the development of various computational tools for characterizing, designing, and optimizing mAb-based therapeutics. The generation and continued pursuit of mAb structural data for in silico analysis has led to the development of several integral databases, notably IMGT^®^, serving as a key resource for data mining [144,145,146]. Molecular dynamic (MD) simulation analysis has driven the computational analysis of the discovery and characterization of relevant molecular interactions within the mAb molecule, mAb binding to biological targets and mAb surface association with the surrounding environment. These interactions can infer intrinsic stability, target binding associations, solubility and aggregation propensity of the mAb. MD simulations and free energy calculations from crystal structures remain key for the highly specific elucidation of mAb-target intermolecular interactions that correlate to binding affinity, as significant interactions such as hydrogen-bond formation can be predicted and determine the strength of the molecular associations [147]. However, elucidation of mAb self-association, solubility, and aggregation propensity have been driven by the development of computational modelling and simulation tools that simultaneously analyses mAb topography and surface polarity. Notably, AGGRESCAN3D, TANGO, and PASTA are the most prominent tools for predicting the site specific aggregation propensity of mAbs [35,148].

The preliminary elucidation of mAb structural data—that being amino acid sequences and higher order structure (HOS)—is experimentally derived to produce crystal structures that model the solid-state 3D structure of the mAb. Mass spectrometry technologies have come of age to produce high throughput and orthogonal analysis to elucidate mAb peptide sequences, oxidation, deamidation, and glycosylation heterogeneity, as well as, more recently, for native, destabilized, and aggregated HOS elucidation [149,150]. X-ray crystallography technologies have been the underlying workhorse for elucidating the crystal structure of mAbs. Producing crystalline mAbs is challenging, owing to the complexity of mAb HOS and the degree of mAb conformational heterogeneity. However, several complementing technologies have evolved in recent years to ascertain structure and interactions, including circular dichroism (CD), infrared (IR) and raman spectroscopy, cryogenic-electron microscopy, and nuclear magnetic resonance (NMR) spectroscopy [148,150,151,152,153,154,155,156,157]. Of the technologies available for HOS elucidation, 2D-NMR and X-Ray crystallography (followed by MS) provide the highest sensitivity and local specificity. In comparison, CD, IR, and raman are high-throughput methods, although they have much lower sensitivity [150,157].

Thus far, only four whole IgG antibodies have a successfully determined crystal structure (PDB ID: 1HZH, 1IGT, 1IGY, and 5DK3), notably 1HZH as a wholly human IgG1 and 5DK3 as a humanized IgG4/κ, both of which have been used as model structures for MD simulations [158]. However, fragment mAbs yield much higher success with producing crystal structures, which has led to much pursuit and contribution in generating mAb fragments and targetting complexing crystal structure libraries for data mining and in silico analysis [159,160].

Upon obtaining relevant crystal structures for a candidate mAb, detailed crystal structure and MD simulation analysis has been extensively used to elucidate the mAbs intramolecular interactions that confer intrinsic stability and intermolecular interactions to confer interactions to biological targets and self-association. The particular binding interface of interest is analyzed, and amino acids in the mAb structure are identified for substitution that can disrupt molecular interactions in the interface in the instance of diminishing target binding and identify potentially unutilized bonding sites and polarity mismatches in the interface that can be improved in the instance of enhancing target binding. MD simulations are then carried out for the native and substituted mAb structures to elucidate any relevant bond formations, interactions, and more favorable binding free energy produced by the substituted structure, of which promising candidates can be synthesized and validated through in vitro analysis. This predictive approach has been successfully applied to identify mutations in mAbs for modulating effector functions, target binding, optimizing affinity capture for manufacturing, and, more recently, for designing antibodies de novo from targets of interest [11,161,162,163,164,165]. Mutations have been specifically identified for enhanced binding to Fc_γ_R2A, Fc_γ_R3A, and Cq1, as well as a reduced binding to Fc_γ_R2B for a more pronounced ADCC and CDC effect, an enhanced binding to FcRn for an extended half-life, and a reduced binding to Fc_γ_Rs and Cq1 for a diminished ADCC and CDC effect, all of which are transferrable to mAbs of the same isotype sub-class [137,141,142]. Further to this, the molecular interactions of glycoengineered mAb variants to FcRs have been characterized through MD simulations to corroborate predictions with the observed modulated effector functions [13,15,137,141,142].

Self-association, solubility, and aggregation propensity are further evaluated by computational modelling tools that specifically characterize the topography and surface polarity of mAbs, concurrently analyzing the local spatial arrangement of amino acids in the mAb structure, solvent accessibility, and local surface charge [5]. In particular, surface exposed hydrophobicity is sought as the lead mechanism for protein self-association driving aggregation propensity, in which aggregation-prone regions (APRs) are identified in the mAb structure. The substitution of identified APRs to gatekeeper (i.e., polar) amino acids has experimentally demonstrated resistance to aggregation and improved solubility, which validates this strategy for improving mAb stability [36,37,38,39,40,166,167,168,169,170]. The majority of APRs identified are located in biologically relevant mAb regions—those being the CDRs and the Fc. Specifically targeting these APRs through mutation may disrupt important biological functions and mAb structure. Analyses of the mAbs intermolecular interactions are necessary to validate the intrinsic stability of substituted mAb variants; if conformational fluctuations are elucidated, then the biologically active interface is likely disrupted. Interestingly, mutations for enhancing biological functions have demonstrated a negative impact on mAb solubility and stability, further suggesting that the self-association profile of mAbs is linked to its biological activity [36,128,171].

Aside from direct residue substitution in mAb structures, other strategies reported to profoundly interfere with the effects of APRs include isotype switching and the strategic addition of N-linked glycans [39,128,172,173]. Additionally, N-linked glycosylation in the CDR regions of mAbs, although uncommon, has demonstrated improved solubility and stability profiles of mAbs without impacting their target affinity, as compared to the same mAbs with removed CDR glycans [128,174]. The rationale behind strategic glycan addition is based on the election of N-linked glycosylation sites that have apparent spacial proximity to APRs, with the introduced glycan therefore sterically hindering the APR from self-association interactions. The benefits of extending mAb half-life with strategic glycan addition, as seen with hyper-glycosylated biotherapeutics (as well as improving mAb solubility and stability for improved formulation strategies) has yet to be realized and suggests a very intriguing and highly relevant technology for perusal.

## 8. Concluding Remarks

Therapeutic antibodies have come of age as continuing to be a key, dominant technology in the biopharmaceutical industry. The repurpose of antibodies to many formats have made them versatile to design and tailor highly specialized treatments, including ADCs as a targeted drug delivery system, bispecific and fragment mAb platforms for tailored engagement and increased bioavailability, and recombinant Fc-fusion proteins for an increased half-life and introduced immunological engagement. Further advancements include the modulation of Fc effector functions through manipulations of the Fc, either through isotype switching, glycoengineering, or strategic mutations in the Fc region, along with the PEGylation of fragment mAbs for enhanced half-life. Advancements in discovery, manufacture, and formulation technologies have further propelled the success of therapeutic antibodies, notably through expression system development and the transition from IV to SC formulations. Human-based expression systems have been extensively used in mAb development and are becoming an accepted manufacturing platform for mAb therapeutics. Furthermore, cell-free synthesis technology is giving rise to the potential for higher efficiency in the manufacture process.

Though developments for further generation antibody therapies have led to great strides in producing improved therapeutic outcomes, many facets of the manufacture process and formulation development strategy pose as challenges to be considered. Antibody-based therapies are susceptible to chemical and enzymatic degradation through oral, nasal, or pulmonary routes of administration and are therefore currently restricted IV or SC delivery. Despite achieving maximum bioavailability through IV/SC administration, tissue penetration of mAb-based therapies is poor, which limits their local bioavailability, requiring high concentrations to achieve an effective dose. The stability of mAbs-based therapeutics is a highly pronounced and recurring challenge to be considered, as it affects manufacture yield and formulation considerations. Several strategies are in development to improve the stability of mAbs in order to potentially produce formulations for pulmonary or oral administration. Notably, computational tools have come of age, complementing experimental techniques to derive antibody structure and aggregation prediction. Through these methods, the stability and aggregation propensity of mAb-based therapies have demonstrated improvement through rational mutation and glycosylation within the framework region, which is potentially translatable to all mAbs within the same isotype. Furthermore, nanocarrier technologies have been shown to enhance the stability and potentially control the release of mAbs. The refinement of rational mAb design coupled with nanocarrier technologies has the potential to overcome these challenges, to develop superior treatment strategies, and ultimately to formulate for non-invasive administration routes such as pulmonary delivery.

## Figures and Tables

**Figure 1 antibodies-08-00036-f001:**
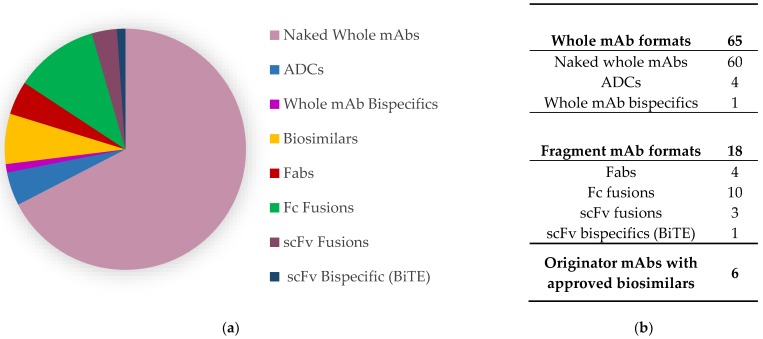
The proportions of therapeutic antibody formats approved for therapeutic use as of December 2018^,^ IMGT^®^ depicted through (**a**) a pie chart and (**b**) a table format.

**Figure 2 antibodies-08-00036-f002:**
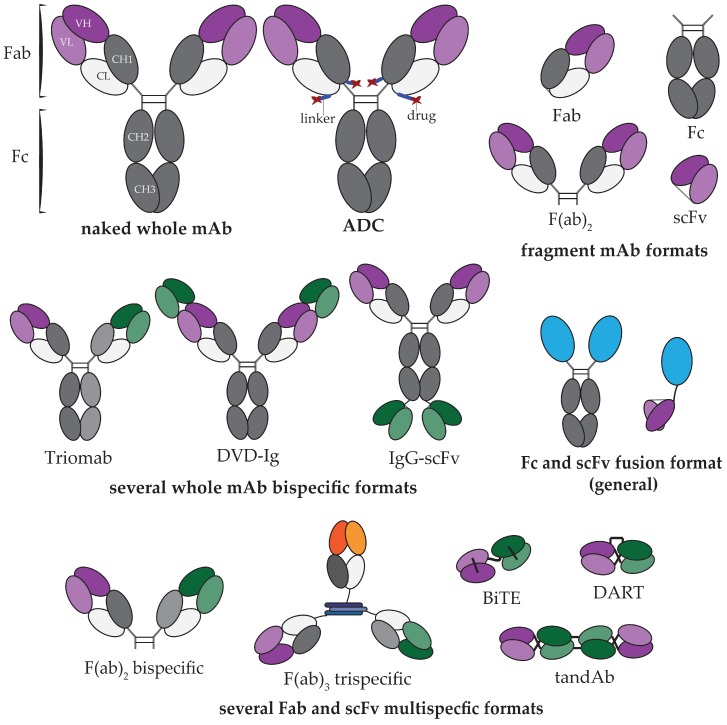
Schematic representation of a whole monoclonal antibody (mAb), a fragment mAb, and prominent fusion mAb formats that have been developed for strategic therapeutic uses. Proteins fused to mAb fragments are depicted as blue ovals for a general representation; however, fusion proteins may vary in size and structure. Fragment formats include the crystallizable (Fc), antigen binding (Fab and F(ab)2), and single-chain variable (scFv) fragments. Further whole mAb formats include the antibody–drug conjugate (ADC), triomab, dual variable domain immunoglobulin (DVD-Ig), and immunoglobulin–scFv fusion (IgG-scFv). Multispecific fragment formats include the F(ab)2 bispecific, bispecific T-cell engager (BiTE), dual affinity re-targeting molecule (DART), and tandem diabody (tandAb).

**Figure 3 antibodies-08-00036-f003:**
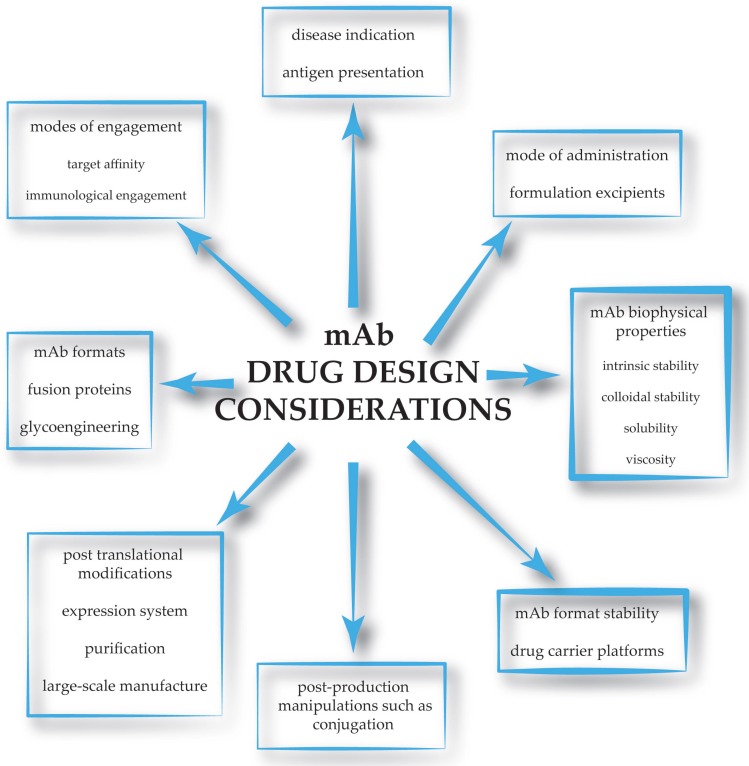
Schematic representation of the concept stages of mAb drug development in which considerations follow on from one process to the next in the design and manufacture of mAb-based therapeutics.

**Table 1 antibodies-08-00036-t001:** Summary of key technological advancements that address challenges and considerations in mAb design, manufacture, and formulation strategies.

Challenges	Advancements
**Manufacture Considerations**	
Hybridoma technology produces immunogenic mAbs	Humanization technologies [4,5,6]
Yield from hybridoma technology is variable	Commercial cell line development and recombinant technology [4,5,6]; lower organism systems for fragment mAbs, mammalian systems for whole mAbs, and Fc fusions [16,30,31]
Significance of post-translational modifications and higher-order structure in mAb product	
CHO expressed mAbs contain an immunogenic glycosylation profile	Human-based expression systems [16,30,31,32]
HEK 293 expressed mAbs are prone to aggregation	HKB-11 and PER.C6 cell lines [32]
Undesirable byproducts produced in the manufacture process	in vitro cell-free synthesis technology [33,34]
Stability of mAb affects manufacture yield due to product loss through aggregation in downstream processing steps	Enhancing mAb stability through framework mutations and hyperglycosylation [5,35,36,37,38,39,40]
**Treatment Considerations for Drug Design and Formulation**	
Susceptibility of mAb to degradation limits delivery to intravenous and subcutaneous only	Enhancing mAb stability through framework mutations, hyperglycosylation [5,35,36,37,38,39,40], and nanocarrier technologies [8,41]
Stability of mAb limits concentration of formulation	
Concentration of mAb affects viscosity and injection pressure for subcutaneous delivery	Excipients, fragment mAbs, PEGylation, and hyperglycosylation [25,26,27,41]
Poor tissue penetration and biodistribution	Fragment mAbs and nanocarrier technologies [8,41,42]; high affinity ADCs [43,44,45]
Reduced half-life in low MW species	PEGylation, hyper-glycosylation and Fc fusion proteins [8,41,46]; modulation of FcRn recycling through Fc mutation [11,13]
Modulation of immunological engagement	Isotype switching, glycoengineering, and Fc mutations [11,13,15,43,46]

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
