# Peer review of "Current Advancements in Addressing Key Challenges of Therapeutic Antibody Design, Manufacture, and Formulation"

_2073-4468, 2019, doi:10.3390/antib8020036_

Reviewer 1 Report

The review submitted by Sifniotis et al. is a very nice summary of past and current practices associated with the manufacturing and discovery of antibody therapeutics. In general, the information is nicely presented and explained and should be of interest to a broad audience due to the introductory nature of the prose (i.e. I would find this suitable for lay readers outside of the field or those just entering the field). There are several minor issues noted below that should be addressed and that will further improve the quality of this manuscript.

General comments:

Several sentences seem rather long and contain multiple/interchanging conclusions. I would recommend reviewing the manuscript and splitting some of these sentences into two, allowing for a focus on single key conclusions and helping guide the reader. I have provided a few example lines in the 'specific comments' below. 

In general, it would be beneficial to provide a summary conclusion within each section. There is a lot of good information presented and gives a good review of the current techniques being utilized. But as a review, the authors should aggregate this information and provide a though on 'best practices' and/or comment on the challenges and limitations in the field that must be overcome. This is done quite well in some section, but please review to make consistent throughout. 

I feel that stating a sentence with the abbreviation 'mAb' isn't correct from prior manuscript edits I have seen in the past and from other reviewers within this same journal. You can either spell out the work (i.e. mAb -> Monoclonal antibodies) or add a transition word to offset this usually (i.e. mAb -> Often mAbs).

Specific comments:

Line 38-40: This sentence is difficult to read and would recommend splitting into two sentences.

Line 58: I don;t think "Development" is the appropriate term for this section, as you focus on the synthesis and production of protein therapeutics. Other suggestions would be: production, fabrication, manufacture. 

Line 68-71: There are those trying to utilize this intra-cellular machinery in vitro to create a mimic the cell. You touch on some aspects alter in this review, but in context of this section, it would be good to comment on specifically why these methods have failed/been difficult (e.g. lack of functional 'machinery' being produced, cost, inconsistency in final result, etc.). This is very important to the overall clinical disposition and consistency of the final product. It would be of interest to readers to know why this method is also not working. Synthetic heparin is a good good case study, but some literature does exist for other protein/mAb therapeutics. 

Line 130-132: This sentence is difficult to read and would recommend splitting into two sentences.

Line 159-163: The thought/goal seems unfinished. You say that Ecoli systems should not be used, but then end with an example where they are. A conclusionary sentence to denote importance or key conclusion would be helpful. 

Line 267-271: I feel that the benefits of nanotechnolgies for delivery to the lung is being glossed over here. A significant level of literature exists (and you can even reference to it versus going in depth yourself). It should also be mentioned any co-formulation issues that arise due to using a mAb either as a payload versus an attached surface molecule. Considering a closing remark is focused on nano-technology and mAb delivery to hard to each tissues, this area should be provided additional information. 

Line 287: There is a reason many common excipients are not used in therapies due to know risks/reactions. Comment on this and make it known.  

Line 313: Instead of 'local delivery', so you mean intra-tumoral, or just tumor targeting? Each has very different logistical issues, implications, and challenges to overcome. 

Section 6 (Lines 345-377): Very heavy subject and lots of information missing. However, for the intent of the review, i agree not all of this information can/should be provided. I suggest including suggested reviews at the end of each paragraph/topic so readers can obtain the necessary details and specifics to define context outside of this review. 

Line 350: Isotype will also affect half-life...so there is a balance of isotype to target to toxicities, as well. Several examples exist where effector response is attempted to be balanced by using faster-cleared isotypes.

Line 400-405: This sentence is difficult to read and would recommend splitting into two sentences.

Line 400-405: Many are trying to get away from crystallization, or using it on a near-final product -- not in the discovery and development stages. What are the pros vs. cons of different methods for structural elevation (e.g. intact antibody MS evaluation). What are the authors recommendations for a 'comprehensive structural analysis'? What needs/should be accounted for? Additional commentary on 2D-NMR use in the review process would also be appreciated as NIST has been using this more and more compared to crystallography. 

Line 424-427: I think the references might be missing form this sentence. 

Line 487-488: I think it is important to note this is focusing on the stability/formulation. I don't think the pharmacologic (or change in pharmacologic) disposition has yet been evaluated and proved superior efficacy in any of the cited literature. A lot of cell line and in vitro work exists, but the in vivo data shows mixed results, highlighting the complexity of combining two highly variable formulations (i.e. antibodies and nanoparticles). 

Section 8: Also should comment that these improved manufacturing/discovery techniques can help understand the variable pharmacology of these agents - hopefully by reducing formulation heterogeneity or reducing non-specific/off-target effects. 

Author Response

“Further whole mAb formats include antibody-drug conjugates (ADCs) that have exceptional potency, isotype switched and glyco-engineered whole mAbs for modulated Fc effector functions and bispecific mAbs with engagement to dual biological targets.” has been amended to “Further whole mAb formats include antibody-drug conjugates (ADCs), bispecifics, isotype switched and glyco-engineered. These additional formats have been strategically designed to introduce exceptional potency, to engage dual biological targets and modulate Fc effector functions.”

“Pharmaceutical Development” was replaced with “mAb Production”, with the title now reading as “2. Overview of mAb Production Challenges and Considerations”.

I believe that the section referenced by the reviewer is not the right place to specifically make mention of the cell-free synthesis technology. The manuscript dedicates paragraph 5 of section 3 to discussion of this technology. However, in this paragraph the following was added to discuss reasons behind the lack of success of this technology thus far:

“Despite the apparent advantages of this technology for mAb manufacture, cell lysates encounter the same challenges as their host cell expression system. That is, post-translational modifications of mAb product from lysates from lower level organisms are limited by the endogenous machinery of that host organism. Furthermore, preparation of mammalian cell lysates is challenging and yield produced from these lysates is considerably low (ref).” which includes new references:

Jaroentomeechai, T.; Stark, J.C.; Natarajan, A.; Glasscock, C.J.; Yates, L.E.; Hsu, K.J.; Mrksich, M.; Jewett, M.C.; DeLisa, M.P. Single-pot glycoprotein biosynthesis using a cell-free transcription-translation system enriched with glycosylation machinery. Nature communications 2018, 9, 2686.

Gurramkonda, C.; Rao, A.; Borhani, S.; Pilli, M.; Deldari, S.; Ge, X.; Pezeshk, N.; Han, T.-C.; Tolosa, M.; Kostov, Y., et al. Improving the recombinant human erythropoietin glycosylation using microsome supplementation in cho cell-free system. Biotechnol. Bioeng. 2018, 115, 1253-1264.

Expression from hybridoma cell line to cell line can vary quite substantially, so to meet the commercial need for high expression efficiency, scalability, quality and reproducibility, high yielding mammalian expression systems have been developed.” has been amended to “Expression, yield and quality of mAb can vary quite substantially in hybridoma cell lines. High yielding mammalian expression systems have been developed to meet the commercial need for high expression efficiency, scalability, quality and reproducibility.”

“Expression platforms that use E. coli in particular are considered high risk due to potential endotoxin contamination in the mAb product, however several approved mAb fragment and recombinant based therapies are produced in an E. coli based expression system including pegol conjugated Fab’ certolizumab pegol (Cimzia®), Fab ranibizumab (Lucentis®) and recombinant Fc fusion romiplostim (Nplate®).” was amended to “Expression platforms that use E. coli in particular are considered high risk due to potential endotoxin contamination in the mAb product, of which complete endotoxin removal requires further purification steps (ref). However, several approved mAb fragment and recombinant based therapies are produced in an E. coli based expression system, including pegol conjugated Fab’ certolizumab pegol (Cimzia®), Fab ranibizumab (Lucentis®) and recombinant Fc fusion romiplostim (Nplate®).” with the following reference:

Pieracci, J.P.; Armando, J.W.; Westoby, M.; Thommes, J. Chapter 9 - industry review of cell separation and product harvesting methods. In Biopharmaceutical processing, Jagschies, G.; Lindskog, E.; Łącki, K.; Galliher, P., Eds. Elsevier: 2018; pp 165-206.

“Lower organism expression platforms such as E. coli and P. pastoris continue to be the preferred option for the manufacture of fragment mAb formats due to the relative ease, high yield and reduced cost for manufacture as compared to mammalian expression platforms.” was added.

Further references were included in this section:

Anselmo, A.C.; Gokarn, Y.; Mitragotri, S. Non-invasive delivery strategies for biologics. Nature Reviews Drug Discovery 2018, 18, 19.

Sousa, D.; Ferreira, D.; Rodrigues, J.L.; Rodrigues, L.R. Chapter 14 - nanotechnology in targeted drug delivery and therapeutics. In Applications of targeted nano drugs and delivery systems, Mohapatra, S.S.; Ranjan, S.; Dasgupta, N.; Mishra, R.K.; Thomas, S., Eds. Elsevier: 2019; pp 357-409.

K Jani, R.; Krupa, G.; Rupal, J. Active targeting of nanoparticles: An innovative technology for drug delivery in cancer therapeutics. Journal of Drug Delivery and Therapeutics 2019.

Abdelaziz, H.M.; Gaber, M.; Abd-Elwakil, M.M.; Mabrouk, M.T.; Elgohary, M.M.; Kamel, N.M.; Kabary, D.M.; Freag, M.S.; Samaha, M.W.; Mortada, S.M., et al. Inhalable particulate drug delivery systems for lung cancer therapy: Nanoparticles, microparticles, nanocomposites and nanoaggregates. J. Controlled Release 2018, 269, 374-392.

“Excipients are considered for a formulation based on their physicochemical properties, pharmacokinetics and safety. For example, polysorbates are commonly used in biologics as a stabilizing agent; however, their addition in high concentration can denature proteins and cause adverse side effects such as injection site reactions (ref).” was added, which includes new references:

Crommelin, D.J.A.; Hawe, A.; Jiskoot, W. Formulation of biologics including biopharmaceutical considerations. In Pharmaceutical biotechnology: Fundamentals and applications, Crommelin, D.J.A.; Sindelar, R.D.; Meibohm, B., Eds. Springer International Publishing: Cham, 2019; pp 83-103.

Singh, S.K.; Mahler, H.-C.; Hartman, C.; Stark, C.A. Are injection site reactions in monoclonal antibody therapies caused by polysorbate excipient degradants? J. Pharm. Sci. 2018, 107, 2735-2741.

Rayaprolu, B.M.; Strawser, J.J.; Anyarambhatla, G. Excipients in parenteral formulations: Selection considerations and effective utilization with small molecules and biologics. Drug Dev. Ind. Pharm. 2018, 44, 1565-1571.

“local” was removed so that the sentence now reads as “Antibody therapies are currently restricted to invasive parenteral routes of administration for maximum bioavailability and systemic distribution, however delivery to the specific target tissue such as tumors for cancer treatments, continues to be a challenge.”

In this instance, both tumor targeting and penetration are implied, as they are discussed as separate challenges through the continued text. The text may not have specifically been clear in distinguishing between tumor targeting and penetration; however, the discussion was extensive and thorough in identifying the different challenges for both.

Firstly, with tumor targeting: “Penetration to tissue from blood vessels is again poor, owing to mAbs polar surface charge and relatively large MW limiting transport through physiological barriers and efficiency of tissue penetration is further influenced by systemic and local mAb clearance rates. Enhancement to mAbs affinity to their biological target through maturation and strategic mutation technologies have led to faster diffusion rates of mAb to target for increased efficacy.”

Then for tumor penetration: “However for tumourous tissue specifically, penetration through tumors is restricted by the tumor’s binding site barrier, in which antigens expressed on the tumors periphery capture the majority of mAb released to surrounding tissue and this effect is exaggerated with overexpressed antigen. Enhancement of whole mAbs affinity beyond 1 nM has specifically shown that there is no further improvement of tumor diffusion rates and tissue penetration and accumulation (ref). Increased affinity of mAbs to cellular targets also leads to increased uptake, internalization and catabolism of mAbs which reduces ADCC and increases mAb clearance rate. This mechanism is exploited through the development of high affinity ADCs to target the delivery and release of potent drugs, inducing targeted cell death (ref).” … with further discussions etc

Several key reviews are cited in this section of the manuscript, notably:

Wang, X.; Mathieu, M.; Brezski, R.J. IgG Fc engineering to modulate antibody effector functions. Protein Cell 2018, 9, 63-73.

Pawlowski, J.W.; Bajardi-Taccioli, A.; Houde, D.; Feschenko, M.; Carlage, T.; Kaltashov, I.A. Influence of glycan modification on IgG1 biochemical and biophysical properties. J. Pharm. Biomed. Anal. 2018, 151, 133-144. 523

Fonseca, M.H.G.; Furtado, G.P.; Bezerra, M.R.L.; Pontes, L.Q.; Fernandes, C.F.C. Boosting half-life and effector functions of therapeutic antibodies by Fc-engineering: An interaction-function review. Int. J. Biol. Macromol. 2018, 119, 306-311.

One further reference was added which summarises current reviews on the topic:

Lei, C.; Gong, R.; Ying, T. Editorial: Antibody Fc engineering: Towards better therapeutics. Front. Immunol. 2018, 9.

A summary sentence for the reader’s referral was added, which includes the aforementioned reviews:

“For a more comprehensive understanding of mAb engineering strategies for modulated immune effector functions, we direct the reader to the following reviews (ref).”

“IgG3 does not efficiently bind to FcRn which reduces its half-life to approximately 7 days, as opposed to a 21 day half-life for the other IgG isotypes. In most instances of mAb design, an extended half-life is preferred as to prolong the effective dose of mAb in serum (ref).” was added, referencing previously referenced review:

Fonseca, M.H.G.; Furtado, G.P.; Bezerra, M.R.L.; Pontes, L.Q.; Fernandes, C.F.C. Boosting half-life and effector functions of therapeutic antibodies by fc-engineering: An interaction-function review. Int. J. Biol. Macromol. 2018, 119, 306-311.

“X-ray crystallography technologies have been the underlying workhorse for elucidating the crystal structure of mAbs, although challenging owing to the difficulties in producing crystalline mAbs, the degree of conformational heterogeneity and HOS complexity associated with the mAbs large molecular structure, however cryogenic-electron microscopy and nuclear magnetic resonance spectroscopy have evolved as a complementing technologies to ascertain structure and interactions in recent years (ref).” has been amended to “X-ray crystallography technologies have been the underlying workhorse for elucidating the crystal structure of mAbs. Producing crystalline mAbs is challenging owing to the complexity of mAb HOS and the degree of mAb conformational heterogeneity. However, several complementing technologies have evolved in recent years to ascertain structure and interactions, including circular dichroism (CD), infrared (IR) and raman spectroscopy, cryogenic-electron microscopy and nuclear magnetic resonance (NMR) spectroscopy (ref).” with the addition of these references:

Blaffert, J.; Haeri, H.H.; Blech, M.; Hinderberger, D.; Garidel, P. Spectroscopic methods for assessing the molecular origins of macroscopic solution properties of highly concentrated liquid protein solutions. Anal. Biochem. 2018, 561-562, 70-88.

Brinson, R.G.; Marino, J.P.; Delaglio, F.; Arbogast, L.W.; Evans, R.M.; Kearsley, A.; Gingras, G.; Ghasriani, H.; Aubin, Y.; Pierens, G.K., et al. Enabling adoption of 2d-nmr for the higher order structure assessment of monoclonal antibody therapeutics. mAbs 2019, 11, 94-105.

Young, J.A.; Gabrielson, J.P. Higher order structure methods for similarity assessment. In Biosimilars: Regulatory, clinical, and biopharmaceutical development, Gutka, H.J.; Yang, H.; Kakar, S., Eds. Springer International Publishing: Cham, 2018; pp 321-337.

Wang, X.; An, Z.; Luo, W.; Xia, N.; Zhao, Q. Molecular and functional analysis of monoclonal antibodies in support of biologics development. Protein & Cell 2018, 9, 74-85.

The text specifically mentions complementing technologies including MS and NMR as such:

“Mass spectrometry (MS) technologies have come of age, to produce high throughput and orthogonal analysis to elucidate mAb peptide sequences, oxidation, deamidation and glycosylation heterogeneity, and more recently for native, destabilized and aggregated HOS elucidation (ref).”

“However, several complementing technologies have evolved in recent years to ascertain structure and interactions, including circular dichroism (CD), infrared (IR) and raman spectroscopy, cryogenic-electron microscopy and nuclear magnetic resonance (NMR) spectroscopy (ref).”

A comparison of the different methods was added, with the following references:

“Of the technologies available for HOS elucidation, 2D-NMR and X-Ray crystallography provide the highest sensitivity and local specificity, followed by MS. In comparison, CD, IR and raman are high-throughput methods although they have much lower sensitivity (ref).”

Young, J.A.; Gabrielson, J.P. Higher order structure methods for similarity assessment. In Biosimilars: Regulatory, clinical, and biopharmaceutical development, Gutka, H.J.; Yang, H.; Kakar, S., Eds. Springer International Publishing: Cham, 2018; pp 321-337

Wang, X.; An, Z.; Luo, W.; Xia, N.; Zhao, Q. Molecular and functional analysis of monoclonal antibodies in support of biologics development. Protein & Cell 2018, 9, 74-85.

References were added (these references also appear with the sentence following, as were originally grouped together to address both ):

Saxena, A.; Wu, D. Advances in therapeutic fc engineering – modulation of igg-associated effector functions and serum half-life. Front. Immunol. 2016, 7.

Booth, B.J.; Ramakrishnan, B.; Narayan, K.; Wollacott, A.M.; Babcock, G.J.; Shriver, Z.; Viswanathan, K. Extending human igg half-life using structure-guided design. mAbs 2018, 10, 1098-1110.

Kellner, C.; Otte, A.; Cappuzzello, E.; Klausz, K.; Peipp, M. Modulating cytotoxic effector functions by fc engineering to improve cancer therapy. Transfus. Med. Hemoth. 2017, 44, 327-336.

“potentially” was added, so the sentence now reads as “Furthermore, nanocarrier technologies have been shown to enhance the stability and potentially control the release of mAbs”

As this wasn’t discussed prior, I believe that it is inappropriate to raise this additional comment in the concluding remarks. This is an ambiguous statement that requires discussion in order to make a concluding remark. To discuss this would require a new section that discusses what causes deviations in pharmacology of mAbs that is affected through their development, production and formulation (with relevant referencing). For the scope of this manuscript, this discussion is omitted. However, a sentence to mention that it is of significance without further discussion is warranted, although not in the concluding remarks section. The following was included in section 2 by separating the second paragraph:

“Many factors through the manufacture process influence glycosylation and charge heterogeneity of mAbs, which affects their biophysical and pharmacological properties. Although not specifically discussed in this review, improvement and control mAb production technologies address these variations to reduce formulation heterogeneity and off-target cytotoxicities.”

Reviewer 2 Report

Dear authors,

This is a nice review paper in the filed of antibody that provides a wide range of knowledge  to develop antibody drugs. Authors might add one section as technology for affinity maturation such as yeast display and error prone PCR etcs.

thanks

Author Response

This seemed appropriate to add to paragraph 2 of section 3, through the body of discussion of phage display technologies. The following was added, with references:

“Error-prone PCR based mutagenesis has propelled display technology for the generation of enormous libraries for target affinity screening. Yeast surface display further improves these screening technologies as glycosylation sites introduced in CDRs are expressed through this platform (ref).”

Frenzel, A.; Roskos, L.; Klakamp, S.; Liang, M.; Arends, R.; Green, L. Antibody affinity. In Handbook of therapeutic antibodies: Second edition, Wiley Blackwell: 2014; Vol. 1-4, pp 115-140.

Elgundi, Z.; Reslan, M.; Cruz, E.; Sifniotis, V.; Kayser, V. The state-of-play and future of antibody therapeutics. Adv. Drug Del. Rev. 2017, 122, 2-19.

Reviewer 3 Report

manuscript is generally well written and provides good introduction into the proposed topic. some minor adjustments are needed and are indicated in the attached document.

Author Response

The figure has been revised and now includes a table listing the values represented in the pie chart as figure 1b. The schematic figure of mAbs has been separated as Figure 2. All figure references have been amended through the body of the text.

The use of the word was deliberate, oligopeptide is defined as a short peptide sequence of 2 – 20 amino acids in length, which can be chemically synthesised.

As thoroughly discussed, if the product allows for a lower organism expression, then it is preferential to develop and take to industrial scale. References added to support the case:

Mizukami, A.; Caron, A.L.; Picanço-Castro, V.; Swiech, K. Platforms for recombinant therapeutic glycoprotein production. In Methods mol. Biol., Humana Press Inc.: 2018; Vol. 1674, pp 1-14.

Behme, S. Manufacturing of pharmaceutical proteins: From technology to economy: Second, revised and expanded edition. wiley: 2015; p 1-427.

“Formulations are optimized to achieve the highest dosing concentration (minimum achievable volume) without compromising the quality of the mAb in formulation (ref).” was amended to ”Formulations are optimized to achieve the highest dosing concentration at the minimum achievable volume for injection, without compromising the quality of the mAb in formulation (ref).”

“Alternative non-invasive administration strategies such as pulmonary delivery causes additional mechanical stress that contribute further to mAb instability and loss, and oral delivery is unsuitable due to chemical degradation and poor absorption in the gastric and intestinal environments (ref).” was amended to “Alternative non-invasive administration strategies such as pulmonary delivery causes additional mechanical stress that contribute further to mAb instability and loss. Furthermore, oral delivery is unsuitable due to chemical and enzymatic degradation, as well as poor absorption in the gastric and intestinal environments (ref)”.

“genes” was replaced with “expressed mAb”, which now reads as “Antibody variable genes from B-lymphocyte pools are isolated through PCR, cloned into a phage expression vector that presents the expressed mAb on the surface of the phage, the library of vectors is rescued in phage and the phage library screened against a specific target antigen.”

“HEK 293 expression systems however are prone to inducing mAb aggregation in culture, which is detrimental to cell viability and loss of product during manufacture, hence HKB-11 and PER.C6 are the preferred commercial human cell line for mAb expression (ref).” was amended to “However, HEK 293 expression systems are prone to inducing mAb aggregation in culture, which is detrimental to cell viability and loss of product during manufacture. Hence, HKB-11 and PER.C6 are the preferred commercial human cell lines for mAb expression in human cell lines specifically (ref).”

The figure has been revised with the following amendments:

“modes of engagement” describes the design considerations for the biophysical properties of the mAb, that being target and immunological engagements. The figure has been revised to clarify the matter with the inclusion of “target affinity” and “immunological engagement”.

The mAb formats have been isolated as a septate point, and further includes “glycoengineering”.

“post translational modifications” has been added to the manufacturing point.

A new point was added in between the drug carrier and formulations point, which includes the following “mAb biophysical properties”, “intrinsic stability”, “colloidal stability”, ‘solubility” and “viscosity”.

“an injectable” was replaced with “a SC formulation” which now reads as “HER2 antibody trastuzumab was originally developed as an IV formulation and was successfully repurposed as a SC formulation (ref); whereas next generation therapies have directly moved towards SC formulation, such as with anti-TNF-α antibody adalimumab (ref).”

“. Notable examples include the TNF Receptor – Fc fusion protein Enbrel® that acts as an inhibitor to overexpressed TNF-α in autoimmune diseases, and further Factor IX – Fc fusion protein Alprolix® that is a blood factor supplement for haemophiliacs” (ref). was added

Section 6 gives a brief overview of the technologies that have advanced mAb therapeutic development through modulation of immune effector functions. Although not directly relating to addressing challenges in mAb manufacture and formulation, this section gives the reader an impression of the developments in mAb glycoengineering and amino acid substitutions. Specifically, the manufacture of glycoengineered mAbs is addressed through the body of section 3, which highlights that these glycoengineered developments require consideration for their production. Furthermore, developments in hyperglycosylation and amino acid substitution are mirrored in the next section, which specifically addresses the rational design of mAbs for enhanced stability. In effect, section 6 is helping to reinforce the potential behind rational design of mAbs, by demonstrating that rational design has thus far correlated with experimental results for the cases presented in modulating immune effector function. Section 6 also gives the reader an impression of the relevance of this field for developing mAb treatment strategies, whilst briefly reporting the significant advancements in this field.

However, to reflect the scope of the manuscript, “Design” was included in the title which now reads as “Current Advancements in Addressing Key Challenges of Therapeutic Antibody Design, Manufacture and Formulation”.

Section 7 delves into the design mAbs for improved target affinity and stability. Notably, it takes an appreciation of the technologies that have generated the necessary structural information for said analysis. However, this discussion directly links to the improved manufacturability and repurpose of mAbs for reformulation through enhancing their intrinsic stability. It is a very significant discussion that contributes to the latest developments in producing highly stable biobetters for improved manufacturability (to increase product yield and quality) and reformulation (to repurpose as highly stable injectables, aerosols etc). It further discusses the potential to repurpose treatment strategies by highlighting hyperglycosylation technology as a means of extending mAb half-life. These advancements appear to have a lot of potential and it was our intention to highlight the rational design technologies in this section as advancements that warrant further development.

As above discussed with the previous comment, to reflect the scope of the manuscript, “Design” was included in the title which now reads as “Current Advancements in Addressing Key Challenges of Therapeutic Antibody Design, Manufacture and Formulation”.
